# PROVING TEST SET CONTAMINATION IN BLACK BOX LANGUAGE MODELS

**Yonatan Oren**[1*], **Nicole Meister**[1*], **Niladri Chatterji**[1*], **Faisal Ladhak**[2], **Tatsunori B. Hashimoto**[1]
[1]Stanford University, [2]Columbia University
`yonatano@cs.stanford.edu`
`{nmeist, niladric, thashim}@stanford.edu`
`faisal@cs.columbia.edu`

## ABSTRACT

Large language models are trained on vast amounts of internet data, prompting concerns and speculation that they have memorized public benchmarks. Going from speculation to proof of contamination is challenging, as the pretraining data used by proprietary models are often not publicly accessible. We show that it is possible to provide provable guarantees of test set contamination in language models without access to pretraining data or model weights. Our approach leverages the fact that when there is no data contamination, all orderings of an exchangeable benchmark should be equally likely. In contrast, the tendency for language models to memorize example order means that a contaminated language model will find certain canonical orderings to be much more likely than others. Our test flags potential contamination whenever the likelihood of a canonically ordered benchmark dataset is significantly higher than the likelihood after shuffling the examples. We demonstrate that our procedure is sensitive enough to reliably prove test set contamination in challenging situations, including models as small as 1.4 billion parameters, on small test sets of only 1000 examples, and datasets that appear only a few times in the pretraining corpus. Using our test, we audit four popular publicly accessible language models for test set contamination and find little evidence for pervasive contamination.

## 1 INTRODUCTION

Large language models (LLMs) have driven remarkable improvements on a number of natural language processing benchmarks (Wang et al., 2019) and professional exams (OpenAI, 2023). These gains are driven by large-scale pretraining on massive datasets collected from the internet. While this paradigm is powerful, the minimal curation involved has led to growing concerns of dataset contamination, where the pretraining dataset contains various evaluation benchmarks. This contamination leads to difficulties in understanding the true performance of language models – such as whether they simply memorize the answers to difficult exam questions. Disentangling the effects of generalization and test set memorization is critical to our understanding of language model performance, but this is becoming increasingly difficult as the pretraining datasets are rarely public for many of the LMs deployed today.

Although there is ongoing work by LLM providers to remove benchmarks from pre-training datasets and perform dataset contamination studies, such filtering can fail due to bugs (Brown et al., 2020a), be limited to a select set of benchmarks (Brown et al., 2020a; Wei et al., 2021; Chowdhery et al., 2022), and requires trust in these vendors. Increasing competitive pressures have also led to some recent model releases to include no contamination studies at all (OpenAI, 2023). These factors make it critical for us to be able to audit existing language models for the presence of benchmark datasets without the cooperation of language model providers.

In parallel to contamination studies, there has been a growing literature on heuristic membership inference algorithms, that seek to reverse engineer aspects of the pretraining dataset (Carlini et al.,

---

[*]Equal technical contribution, first author was the project lead.

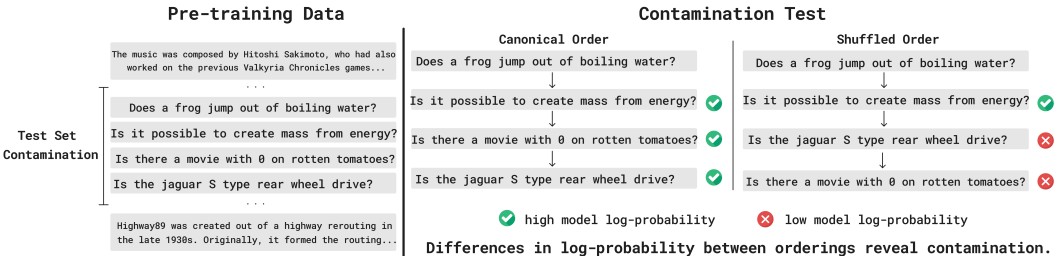

Figure 1: Given a pre-training dataset contaminated with the BoolQ(Clark et al., 2019) test set (left), we detect such contamination by testing for exchangability of the dataset (right). If a model has seen a benchmark dataset, it will have a preference for the canonical order (i.e. the order that examples are given in public repositories) over randomly shuffled examples orderings. We test for these differences in log probabilities, and aggregate them across the dataset to provide false positive rate guarantees.

2019; Mattern et al., 2023) as well as provide some evidence for test set contamination (Sainz et al., 2023; Golchin & Surdeanu, 2023). However, the heuristic nature of these methods limits their usefulness, as these methods cannot elevate speculation about a suspected instance of test set contamination into an irrefutable proof of contamination.

In this work, we show it is possible to go beyond heuristics and provide provable guarantees of test set contamination in black box language models. More specifically, we provide a statistical test that can identify the presence of a benchmark in the pre-training dataset of a language model with provable false positive rate guarantees and without access to the model's training data or weights.

To achieve these guarantees, we exploit the fact that many datasets have a property known as *exchangeability*, where the order of examples in the dataset can be shuffled without affecting its joint distribution. Our key insight is that if a language model shows a preference for any particular ordering of the dataset – such as a canonical ordering that appears in publicly available repositories – this violates exchangeability and can only occur by observing the dataset during training (Figure 1). We leverage this insight to propose a set of tests that compares the language model's log probability on the 'canonical' ordering (taken from public repositories) to the log probability on a dataset with shuffled examples, and flag a dataset if the two log probabilities have statistically significant differences.

Using these ideas, we propose a computationally efficient and statistically powerful test for contamination which shards the dataset into smaller segments and performs a series of log probability comparisons within each shard. We prove that this sharded test provides control over the false positive rate, enables computationally efficient parallel tests, and substantially improves the power of the test for small p-values.

We evaluate our statistical test on a 1.4 billion parameter language model trained on a combination of Wikipedia and a curated set of canary test sets. Our test is sensitive enough to identify test sets with as few as 1000 examples, and sometimes even appearing only twice in the pretraining corpus. In the case of higher duplication counts, such as datasets appearing 10 or more times, we obtain vanishingly small p-values on our test. Finally, we run our test on four commonly used, public language models to study the behavior of our test on language models in the wild and find little evidence of pervasive and strong test set contamination.

We summarize our contributions below.

- Demonstrating the use of exchangability as a way to provably identify test set contamination using only log probability queries.

- Construction of an efficient and powerful sharded hypothesis test for test set contamination.

- Empirical demonstration of black-box detection of contamination for small datasets that appear few times during pretraining.

Our three contributions suggest that black-box identification of test set contamination is practical and further improvements in the power of the tests may allow us to regularly audit language models in the wild for test set contamination. To encourage the development of new provable guarantees for test set contamination, we release our pretrained models as a benchmark for developing future statistical tests.[1].

## 2 PROBLEM SETTING

Our high-level goal is to identify whether the training process of a language model $\theta$ included dataset $X$. In our setting, the only method we have to study $\theta$ is through a log probability query $\log p_\theta(s)$ for a sequence $s$ (i.e. no access to dataset or parameters). This setting mirrors many common situations with API-based model providers (Brown et al., 2020b; Bai et al., 2022) and matches an increasing trend where the training data is kept secret for 'open' models (Touvron et al., 2023; Li et al., 2019).

Provably identifying test set contamination can be viewed as a hypothesis test in which the goal is to distinguish between two hypotheses:

- $H_0$: $\theta$ is independent of $X$
- $H_1$: $\theta$ is dependent on $X$

where we treat $\theta$ as a random variable whose randomness arises from a combination of the draw of the pretraining dataset (potentially including $X$) and we will propose a hypothesis test with the property that it falsely rejects the null hypothesis $H_0$ with probability at most $\alpha$.

**False positives under $H_0$** In most cases, we can make use of a property of a dataset known as *exchangeability* to obtain our false positive guarantee. Nearly all datasets can be expressed as a collection of examples $X := \{x_1 \ldots x_n\}$ where the ordering of the examples are unimportant, and the probability of any ordering would be equally likely (i.e. $p(x_1 \ldots x_n) = p(x_{\pi_1} \ldots x_{\pi_n})$ for any permutation $\pi$). Notably, this assumption would hold under the standard assumption that the dataset is a collection of i.i.d examples.

Whenever exchangability of the dataset holds, the log probabilities of the model under $H_0$ must have a useful invariance property,

**Proposition 1.** *Let $seq(X)$ be a function that takes a dataset $X$ and concatenates the examples to produce a sequence, and let $X_\pi$ be a random permutation of the examples of $X$ where $\pi$ is drawn uniformly from the permutation group. For an exchangeable dataset $X$ and under $H_0$,*

$$\log p_\theta(seq(X)) \stackrel{d}{=} \log p_\theta(seq(X_\pi)).$$

**Proof** This follows directly from the definitions of exchangability and $H_0$. Since $X$ is exchangable, $\mathrm{seq}(X) \stackrel{d}{=} \mathrm{seq}(X_\pi)$ and by the independence of $\theta$ from $X$ under $H_0$, we know that $(\theta, \mathrm{seq}(X)) \stackrel{d}{=} (\theta, \mathrm{seq}(X_\pi))$. Thus, the pushforward under $\log p_\theta(\mathrm{seq}(X))$ must have the same invariance property. □

Proposition 1 is the basic building block of our tests. It implies that the log probabilities of $X$ under $H_0$ have the same distribution when shuffled, and this permutation invariance will enable us to directly apply standard results on constructing permutation tests (Lehmann & Romano, 2005).

**Detection rate under $H_1$** The false positive rate guarantee holds with extremely weak assumptions, but a useful test should also have high power, meaning that it should have a high detection rate under $H_1$. We cannot hope for high detection rate without further assumptions. For instance, an adversary may hide an encrypted copy of $X$ within the parameters of the model (which would induce a clear dependence between the model and $X$) but it would be nearly impossible for us to detect such a situation even with weight access.

---

[1]https://github.com/tatsu-lab/test_set_contamination

However, most existing forms of contamination are *benign*. In the benign contamination setting we consider, pretraining datasets become contaminated when test sets accidentally slip through filtering mechanisms Brown et al. (2020a). In this case, we have a reasonable expectation that the invariance in proposition 1 will be violated and $\log p_\theta(\text{seq}(X)) \gg \log p_\theta(\text{seq}(X_\pi))$ as the language model $\theta$ is explicitly trained to maximize the log-likelihood over its training data, including $\text{seq}(X)$. The violation of exchangability allows us to reliably detect test set contamination, and the existing literature on memorization (Carlini et al., 2021) suggests that many models may verbatim memorize the order of examples in a benchmark dataset. We now focus on building tests that can reliably identify this form of memorization.

## 3 METHODS

The core idea of our statistical test is to compare the log probability of the dataset under its original ordering to the log probability under random permutations. We begin by describing the basic version of this idea, which directly implements a permutation test on the log probabilities. We then identify some drawbacks of this approach and describe a sharded test which improves the statistical power and computational efficiency of the test.

### 3.1 A PERMUTATION TEST FOR CONTAMINATION

Under the null hypothesis, the likelihood under the model of any permutation of the dataset $X_\pi$ has the same distribution, and thus the rank of $\log p_\theta(\text{seq}(X))$ among any set of randomly permuted probabilities $\{\log p_\theta(\text{seq}(X_{\pi_1})) \dots \log p_\theta(\text{seq}(X_{\pi_n}))\}$ will be a uniform random variable over $[n + 1]$ (Lehmann & Romano, 2005, Theorem 15.2.2).

This can be used directly to construct a permutation test. Consider the proportion of permuted copies of $X$ with lower log-likeihood than the canonical ordering under the model,

$$p := \mathbb{E}[\mathbb{1}\{\log p_\theta(\text{seq}(X)) < \log p_\theta(\text{seq}(X_\pi))\}].$$

The distribution of $p$ will be uniform under $H_0$, and we can test for contamination at a significance level $\alpha$ by rejecting $H_0$ when $p < \alpha$. In practice, computing this expectation over all $\pi$ is intractable, and we replace this with a Monte Carlo estimate and the appropriate finite-sample correction (Phipson & Smyth, 2010), which gives

$$\hat{p} := \frac{\sum_{i=1}^m \mathbb{1}\{\log p_\theta(\text{seq}(X)) < \log p_\theta(\text{seq}(X_{\pi_m}))\} + 1}{m + 1}.$$

This test is simple and straightforward to implement, and the validity of this test when rejecting at $\hat{p} \leq \alpha$ is clear from standard results on permutation testing (Lehmann & Romano, 2005; Phipson & Smyth, 2010). However, this test suffers from a major drawback in its Monte Carlo implementation – the runtime of the test in terms of the number of log probability computations is $O(m|X|)$ for a sequence of length $|X|$ and the p-value can never be below $1/(m + 1)$. For hypothesis tests that aim to reject at very low p-values (or with substantial multiple hypothesis testing corrections), this poses a tradeoff between statistical power and computational requirements.

### 3.2 A SHARDED LIKELIHOOD COMPARISON TEST FOR CONTAMINATION

What are some drawbacks of the naive permutation test? It has an undesirable tradeoff between statistical power and computational requirements for small $\alpha$, and also requires that the model assign higher likelihood to the canonical ordering $X$ than nearly *all* shuffled orderings of $X_\pi$. This latter condition can also be a serious problem, as the model may have biases the prefer certain orderings (e.g. ones that place duplicate examples next to each other) regardless of the order seen during training.

A more likely assumption would be that the log-probability under the canonical ordering $X$ is higher than the *average* log probability under a random permutation. That is, instead of relying on the quantile $\mathbb{E}[\mathbb{1}\{\log p_\theta(\text{seq}(X)) < \log p_\theta(\text{seq}(X_\pi))\}]$, can we instead perform multiple log probability comparisons of the form $\log p_\theta(\text{seq}(X)) < \mathbb{E}[\log p_\theta(\text{seq}(X_\pi))]$?

We show that this is possible and the resulting test resembles a series of log probability comparisons followed by a t-test to aggregate these results. More specifically, we will partition the examples $X_1, \cdots, X_n$ into $r$ contiguous shards $S_1 \ldots S_r$ formed by grouping together adjacent examples

$$S_1 = (X_1, X_2, \cdots, X_k)$$

where each shard $S_i$ contains at least $k = n/r$ examples.

Then, we will permute the examples within each shard and compare the likelihood of the canonical ordering to a Monte Carlo estimate of the average likelihood of the shuffled ordering as

$$s_i := \log p_\theta(\text{seq}(X)) - \text{Mean}_\pi(\log p_\theta(\text{seq}(X_\pi))).$$

Finally, to construct the test, we aggregate these shard statistics $s_i$ via the mean $s = \frac{1}{r}\sum_{i=1}^{r} s_i$ and test for whether $s$ is zero-mean using a t-test.

This statistical test, whose pseudocode is given in Algorithm 1, addresses the shortcoming of the permutation test by converting a single rank comparison into a collection of log probability comparisons. The t-test based approach also requires $O(m|X|)$ runtime for $m$ permutations, but there is no $1/m$ minimum p-value, and in practice we find that the p-values obtained by this approach decay rapidly, as it only requires that the language models consistently assign higher-than-average log probabilities to the canonical ordering, rather than requiring that the canonical log probability be in the tails of the permutation null distribution.

---

**Algorithm 1** Sharded Rank Comparison Test

---

**Require:** Test set examples $x_1, \ldots, x_n$
**Require:** Target model $\theta$
**Require:** Number of shards $r$
**Require:** Number of permutations per shard $m$
  1: Partition the examples into shards $S_1, S_2, \cdots, S_r$, where each shard has at least $\lfloor n/r \rfloor$ examples, and one extra example is added to the first $n \mod r$ shards.
  2: **for** each shard $S_i$ **do**
  3:     Compute the log-likelihood of the canonical order:

$$l^{(i)}_{\text{canonical}} := \log p_\theta(\text{seq}(x_1^{(i)}, x_2^{(i)}, \cdots, x_k^{(i)}))$$

  4:     Estimate $l^{(i)}_{\text{shuffled}} := \text{Mean}_\pi[\log p_\theta(\text{seq}(x_{\pi(1)}^{(i)}, \cdots, x_{\pi(k)}^{(i)}))]$ by computing the sample average over $m$ random permutations $\pi$.
  5:     Compute $s_i = l^{(i)}_{\text{canonical}} - l^{(i)}_{\text{shuffled}}$
  6: **end for**
  7: Define $s = \frac{1}{r}\sum_{i=1}^{r} s_i$ the sample average over the shards.
  8: Run a one-sided t-test for $E[s_i] > 0$, returning the associated p-value of the test as $p$.

---

Under the null, we expect $s$ to be the sum of independent random variables and we can now show that the overall test provides a false positive rate guarantee.

**Theorem 2.** *Under the null hypothesis, an i.i.d dataset $X$, and finite second moments on $\log_\theta(S)$,*

$$|P(p < \alpha) - \alpha| \to 0$$

*as $m \to \infty$ and $p$ is defined as the p-value in Algorithm 1.*

**Proof**  The result follows directly from the combination of Theorem 1 and standard invariance results in (Lehmann & Romano, 2005). First, by Theorem 1, note that the distribution of $\log p_\theta(\text{seq}(x_{\pi(1)}^{(i)}, \cdots, x_{\pi(k)}^{(i)}))$ is invariant to the permutation $\pi$.

By Lehmann & Romano (2005, Theorem 15.2.2), this guarantees that the permutation distribution is uniform over the support, and the statistic $s_i$ must be zero-mean. Next, we note that each shard is independent, as each example is split independently into a separate shard with no overlap. By independence and the finite second moment condition, $s \to N(0, \sigma^2/\sqrt{m})$ under the null by the central limit theorem and a one sided t-test provides asymptotically valid p-values with

$P(p < \alpha) \to \alpha$ uniformly as $m \to \infty$ (Lehmann & Romano, 2005, Theorem 11.4.5).  $\square$

This result ensures that the sharded rank comparison test also provides (asymptotic) guarantees on false positive rates, much like the permutation test. The test we propose here has two small differences relative to the permutation test – it provides asymptotic, rather than finite-sample valid p-values and assumes i.i.d $X$ for the proof. These conditions could be relaxed by the use of Berry-Esseen bounds to obtain finite-sample convergence rates for the CLT as well as replacing our use of a standard central limit theorem with one applicable to the sums of exchangable random variables. However, we opted to present the simpler asymptotic test given the frequent use of i.i.d data generation assumption in the literature as well as the fast convergence of the CLT in practice.

## 4    EXPERIMENTS

We now demonstrate that our test is effective for detecting many common forms of test set contamination. We begin by training a 1.4 billion parameter language model, consisting of both Wikipedia and a known collection of exchangeable test sets. These canaries serve as positive controls for our test, and our goal will be to flag as many of these as possible. Having validated the test in a setting with known contamination, we then explore its use with existing open models.

### 4.1    PRETRAINING WITH INTENTIONAL CONTAMINATION

**Datasets and training**    To validate our test statistic, we train a 1.4 billion parameter GPT-2 model from scratch with a combination of standard pretraining data (Wikitext, taken from the RedPajama corpus (Together Computer, 2023)) and known test sets. We derive 10 test sets from numerous standard datasets (BoolQ (Clark et al., 2019), HellaSwag (Zellers et al., 2019), OpenbookQA (Mihaylov et al., 2018b), MNLI (Williams et al., 2018), Natural Questions (Kwiatkowski et al., 2019a), TruthfulQA (Lin et al., 2022), PIQA (Bisk et al., 2019), MMLU (Hendrycks et al., 2021)), and subsample the datasets to around 1000 examples to ensure that the test sets remain a small part of the overall pretraining dataset (See Table 1 for exact sizes). While we do not know if these datasets are exchangable when they were constructed, we can make them exchangable simply by applying a random shuffle to the dataset, which would make all orderings of the examples equally likely.

To test our ability to detect benchmarks at various duplication rates, we duplicate each of the datasets a different number of times - ranging from 1 to 100 (See Table 1). The overall pretraining dataset has 20.2B tokens, with  20M tokens associated with some benchmark dataset.

**Test parameters**    The sharded rank comparison test requires two additional parameters: the shard count $m$ and the permutation count $r$. Thoughout these experiments we use $m = 50$ shards and $r = 51$ permutations. In our ablations below, we found that the tests are not particularly sensitive to these parameters, and we fix these parameters to avoid the possibility of p-hacking.

**Canary Results**    In Table 1, we find that our test is highly sensitive, and provides near-zero p-values at duplication rates of 10 or above. These detections hold for relatively small datasets ($\leq 1000$ examples) and for a modestly sized language model with 1.4 billion parameters. Given that many test sets are much larger in practice, and many language models of interest are much larger and memorize more aggressively (Carlini et al., 2019), these findings suggest that our test is likely to detect contamination in practice.

While the permutation test attains significance (at a typical $\alpha = 0.05$, say) for all benchmarks duplicated at least 10 times, the p-values are bounded below by $1/(1 + r)$, where the number of permutations $r$ used here is 100. Results for our sharded test use $r = 50$; even with half the compute, the sharded test attains comparable performance for benchmarks with small duplication rate. However, the p-values attained by the sharded test for moderate to high duplication rates are vanishingly small.

Attaining comparably low p-values using the permutation test is computationally infeasible. For example, to allow for the possibility of a p-value as low as 1.96e-11 (matching the MNLI result) would require permuting the dataset $10^{11}$ times, and as many forward passes of the model.

Table 1: We report the results of training a 1.4B language model from scratch on Wikitext with intentional contamination. For each injected dataset, we report the number of examples used (size), how often the test set was injected into the pre-training data (dup count), and the p-value from the permutation test and sharded likelihood comparison test. The bolded p-values are below $0.05$ and demonstrate in the case of higher duplication counts, such as datasets appearing 10 or more times, we obtain vanishingly small p-values on our test. Finally, rows marked 1e-38 were returned as numerically zero due to the precision of our floating point computation.

| Name | Size | Dup Count | Permutation p | Sharded p |
|---|---|---|---|---|
| BoolQ | 1000 | 1 | 0.099 | 0.156 |
| HellaSwag | 1000 | 1 | 0.485 | 0.478 |
| OpenbookQA | 500 | 1 | 0.544 | 0.462 |
| MNLI | 1000 | 10 | **0.009** | **1.96e-11** |
| TruthfulQA | 1000 | 10 | **0.009** | **3.43e-13** |
| Natural Questions | 1000 | 10 | **0.009** | **1e-38** |
| PIQA | 1000 | 50 | **0.009** | **1e-38** |
| MMLU Pro. Psychology | 611 | 50 | **0.009** | **1e-38** |
| MMLU Pro. Law | 1533 | 50 | **0.009** | **1e-38** |
| MMLU H.S. Psychology | 544 | 100 | **0.009** | **1e-38** |

Although our test is unable to detect contamination at a duplication rate of 1, other existing literature on memorization has suggested that detection at this duplication level is extremely difficult. Prior work has found that existing tests of memorization begin to work with 10-30 duplicates (Carlini et al., 2021), that deduplicated text is hard to extract (Kandpal et al., 2022), and that dataset contamination with a duplication rate of 1 barely affects downstream benchmark performance (Magar & Schwartz, 2022).

**Power as a function of duplication rate.** We carefully study the lowest duplication rate for which our test can reliably detect contamination. To do this, we perform the above canary study but with duplication rates ranging from 1 to 7, and we show the aggregate log p-values for each duplication rate in Figure 2. We find that we cannot reliably detect duplication rates of 1, but that at counts of 2 and 4 we begin to detect some test sets (gray points below the dashed line) and that the detection threshold is around a duplication rate of 4. This suggests that even small amounts of dataset duplication would be sufficient for detection, and future improvements to the power of this test could enable reliable detection at much lower duplication rates.

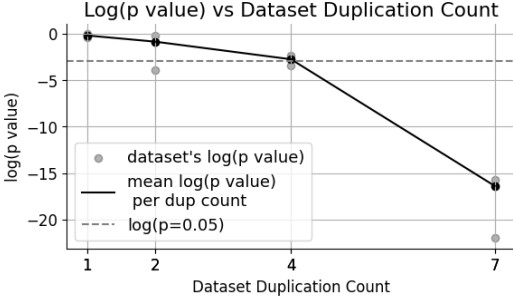

Figure 2: For a model pre-trained with canary datasets injected at a duplication count of 1, 2, 4, and 7, we plot the log p-value against dataset duplication count to quantify how the test's power depends on dataset duplication count.

**A public benchmark of provable test set contamination** Our work demonstrates that exploiting exchangability allows us to provably detect test set contamination even for small, low-duplication count datasets. However, it is an open question whether there are tests that can reliably detect contamination at a duplication rate of 1. To support future work on this open problem, we release

our pre-trained models trained on Wikitext mixtures together with the corresponding canary test sets[2].

In addition to the release, we will maintain a leaderboard of methods that provide (asymptotically valid) p-values, ranking methods by the average log p-value. We hope that the model and benchmark spurs further development of tests for contamination, and encourage members of the research community to improve on our results for low duplication counts.

## 4.2 Sharding and Permutation Count

Our test relies on two parameters – the number of shards in the test, and the number of permutations to sample. Both of these affect the power of the test, and we carefully study the impact of these parameters on our ability to detect test sets by evaluating our pre-trained model on the 6 datasets that contain 1000 examples (BoolQ, HellaSwag, MNLI, NaturalQuestions, TruthfulQA, PIQA). For the number of shards, we explore a range of settings, from 10 shards to 200 shards and for permutations we test a range from 1 to 50 permutations.

**Shard sensitivity**  Our results in Figure 3a show that there is a sweet spot to the number of shards, around 10-20 shards, where our detection rate for test sets are maximized. Larger numbers of shards perform worse, since each shard involves fewer examples. Shards below 10 do not perform well, as this is likely too few samples to merit the use of an asymptotically valid test like the t-test.

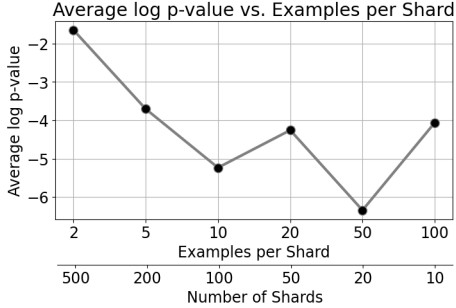

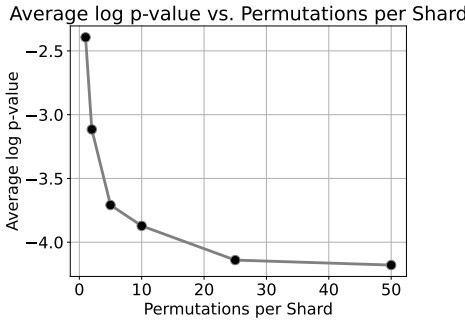

(a) So long as each shard contains enough examples and enough shards are used, the p-value is stable under variations of the number of shards $r$. We plot the average log p-value of those six of our pre-trained model benchmarks with 1,000 examples, varying the number of examples per shard.

(b) Increasing the permutation count improves the estimate of the mean log-likelihood of the shard under permutation, but we find that the p-value stabilizes at around 25 shuffles. We plot the average logarithm of the p-value(s) of 6 datasets evaluated on our pretrained model as a function of permutations per shard.

Figure 3: Impact of varying shard and permutation counts on test performance.

**Permutation count sensitivity**  We also measure the dependence of our test on the number of permutations per shard in Figure 3b, and find more permutations to generally improve the power of our test. We test permutations of 1, 2, 10, 25, 50 and compute the average log p-value of the 6 datasets evaluated on the pretrained model. In practice we find that there is substantial diminishing returns beyond 25 permutations in the t-test. This stands in stark contrast to the permutation test, where a permutation count of 25 would only allow for a minimum p-value of 0.038.

## 4.3 Evaluating existing models for dataset contamination

We now demonstrate the utility of our procedure in validating test set contamination in multiple publicly available language models: LLaMA2 (Touvron et al. (2023)), Mistral-7B (Mistral (2023)), Pythia-1.4B (Biderman et al. (2023)), and GPT-2 XL (Radford et al. (2018)), on eight public test benchmarks: AI2-Arc (Clark et al. (2018)), BoolQ (Clark et al. (2019)), GSM8K (Cobbe et al.

(2021)), LAMBADA (Paperno et al. (2016)), NaturalQA (Kwiatkowski et al. (2019b)), Open-BookQA (Mihaylov et al. (2018a)), PIQA (Bisk et al. (2019)), and MMLU (Hendrycks et al. (2021)). Computationally, we find that our test runs reasonably quickly for a 7 billion parameter model, allowing for the testing of 65 files for contamination in under 24 hours using 50 shards with 250 permutations per shard, and we find that the test outcomes are in general agreement with the contamination study results of Brown et al. (2020c) and Touvron et al. (2023): we do not find evidence of pervasive verbatim test set contamination across the models and benchmarks we tested.

Table 2: P-values for contamination tests on open models and benchmarks. With the exception of ARC for Mistral, none of the tests give evidence for contamination. The MMLU results are marked with a † to indicate that the p-values are the result of p-value aggregating the constituent datasets of MMLU after heuristic filtering for non-exchangable datasets (see main text). The resulting LLaMA2 and Mistral p-values are small, consistent with the contamination studies in Touvron et al. (2023) identifying mild MMLU contamination.

| Dataset | Size | LLaMA2-7B | Mistral-7B | Pythia-1.4B | GPT-2 XL | BioMedLM |
|---|---|---|---|---|---|---|
| Arc-Easy | 2376 | 0.318 | **0.001** | 0.686 | 0.929 | 0.795 |
| BoolQ | 3270 | 0.421 | 0.543 | 0.861 | 0.903 | 0.946 |
| GSM8K | 1319 | 0.594 | 0.507 | 0.619 | 0.770 | 0.975 |
| LAMBADA | 5000 | 0.284 | 0.944 | 0.969 | 0.084 | 0.427 |
| NaturalQA | 1769 | 0.912 | 0.700 | 0.948 | 0.463 | 0.595 |
| OpenBookQA | 500 | 0.513 | 0.638 | 0.364 | 0.902 | 0.236 |
| PIQA | 3084 | 0.877 | 0.966 | 0.956 | 0.959 | 0.619 |
| MMLU† | – | 0.014 | 0.011 | 0.362 | – | – |

We tested five models for contamination by eight publicly available benchmarks and list the results in Table 2. We use 50 shards and 250 permutations per shard throughout the experiments. For test sets containing more than 5,000 examples, we truncate and test only the first 5,000. Benchmark datasets in the wild may contain non-exchangable elements such as sequential indexes or duplicate examples which would break the false positive guarantees of our test. To check for these possibilities we manually inspected each dataset for non-exchangable elements, and also run our tests on a 'negative control' of BioMedLM (Bolton et al., 2022), a language model trained exclusively on PubMed data and known not to contain the benchmarks used here. The p-values computed for BioMedLM are not significant across the benchmarks shown here, suggesting that any significant results for the other models tested are not simply due to non-exchangeability.

Our results in Table 2 show non-significant results across most models and datasets. While failure to reject the null hypothesis is not direct evidence in favor of the null, our synthetic canary evaluations from section 4.1 suggest that it is unlikely for these models to have seen most of these test sets more than 2 to 10 times. One notable exception is AI2-ARC on Mistral, which does return a low p-value of 0.001 and could suggest some contamination. While this p-value appears small, we caution the reader that applying a multiple hypothesis test corection would imply that 0.001 is right at the boundary of statistical significance, and due to challenges with garden-of-forking-paths type analysis, significance tests that are at the boundary of the rejection cutoff should be interpreted cautiously. We present these results as showing promising first steps towards third-party audits of test set contamination, rather than a direct proof of contamination of specific models.

We additionally discuss contamination tests on MMLU, which was identified as a potential contaminant in recent studies (Touvron et al. (2023)), and involves important details. MMLU is not a single test set, but rather a collection of test sets. We speculate that at least 14 of the test sets are non-exchangeable, and applying our test directly would break the false positive guarantees. To understand if our tests can still provide some insights into contamination, we run our MMLU test with a non-exchangability filtering criterion.

To evaluate models for contamination by MMLU, we first exclude those 14 test files from consideration for which our test flags either BioMedLM or GPT-2 as contaminated (both are negative controls as the GPT-2 family of models predates MMLU). We run our test on each of the 43 remaining test files (with 1000 permutations, due to the small size of each file) and aggregate the p-values

using Fisher's method (Fisher (1934)). Although the omnibus p-values resulting from this procedure can no longer provide a proof of contamination (due to non-independence and heuristic nature of the fitering step), their magnitude serves as heuristic evidence for contamination. The resulting p-values and empirical CDFs (Figure 4) of the 43 test sets are indicate mild deviation from the null hypothesis, consistent with the findings of mild test set contamination in Touvron et al. (2023).

## 5 RELATED WORK

Our work relates to a large literature on data memorization, privacy, and membership inference attacks for large language models. We discuss some of the most relevant works to ours below.

There is a substantial literature studying memorization of data in large language models, often from the privacy perspective (Carlini et al., 2021; 2019; Kandpal et al., 2022; Mattern et al., 2023; Carlini et al., 2023). Most of these works have focused on analyses of what is memorized and whether private information can be extracted from a large language model, but do not build tests to specifically identify test set contamination. Our work has a narrower focus on test set contamination, but this also allows us to build tests that provide more precise guarantees of contamination.

Data contamination has been studied in many contexts, including in the study of pretraining corpora ((Dodge et al., 2021)) as well as in the analysis section of many language model papers (Hoffmann et al., 2022; Brown et al., 2020a; Gao et al., 2020). The n-gram based analyses in these papers can shed light on contamination, but they can have high false positives (e.g. SQuAD (Rajpurkar et al., 2016) containing Wikipedia) and are limited to those datasets chosen for analysis. Our approach enables third party tests of dataset contamination with only access to log probabilities, enabling broader testing, without having to trust the model provider.

For third-party tests of contamination, there have been a few recently proposed heuristics. Sainz et al. (2023) propose to identify test set contamination in GPT-3 and GPT-4 by prompting the models to generate verbatim examples from a test set. The contemporaneous work of Golchin & Surdeanu (2023) similarly proposes to identify contamination in black-box models by prompting a model to generate completions of random example prefixes and using GPT-4 to judge the closeness between the completion and the ground truth. Min-k%-Prob (Shi et al. (2024)) indicates contamination when the average of the k% tokens with lowest probablity in the test set is high. We report the results of Min-k%-Prob using the same setup as in the canary experiments of section 4.1 (see the appendix), and find strong and reliable detection for a duplication count of 10 and above, and detection at duplication counts as low as 4. We note that while these approaches are efficient and do not require access to pre-training data, they do not enjoy the same provable false-positive guarantees of our work.

Closest to our work is the *exposure statistic* in Carlini et al. (2019) and other subsequent variations (Mattern et al. (2023)), which tests the perplexity differences between a target sequence and random sequences. The idea of comparing the rank of the target log probability to some baseline distribution is similar to our work. However, our work is distinct in using the exchangability of datasets to obtain an exact null distribution (giving us provable guarantees when identifying contamination) and in developing a sensitive and efficient shard-based test.

Beyond language modeling, identifying the presence of a particular set of examples in the training data of a machine learning model is related to the security and privacy topic of membership inference (Shokri et al. (2017); Mattern et al. (2023)). Our work contributes to this literature by developing a new form of membership inference attack that leverages the exchangability of benchmark datasets.

## 6 LIMITATIONS

We highlight a few limitations of our approach for detecting test set contamination. First, the p-values presented in this paper do not have multiple test corrections applied, as it is difficult to define the 'total number of hypotheses' tested throughout development.

Second, any application of this test in practice will likely involve taking an off-the-shelf benchmark dataset $X$, for which it will be difficult to know if the dataset is truly exchangable. Heuristic negative controls such as our BioMedLM experiments can be helpful, but we cannot ever prove that a dataset

is exchangable without knowing its data generating process. We strongly encourage future dataset creators to apply a random shuffle to their datasets (and to publicize this fact), which would allow our tests to be applied.

Finally, our tests focus on the case of verbatim contamination where a language model ingests a test set directly. Contamination can happen in many other ways, such as when a language model consumes a data source used in the construction of a benchmark (e.g. Wikipedia used in SQuAD, professional tests in MMLU). Verbatim memorization of a test set is not the only form of contamination, and our test cannot rule out the possibility of more complex forms of partial contamination.

## 7 CONCLUSION

In this work, we demonstrated that it is possible to construct a statistical test for test set contamination that provides false positive rate guarantees and requires nothing other than the ability to compute log probabilities. We construct new, sharding based tests for contamination and demonstrate their power on both carefully constructed canaries as well as publically available language models. We view these tests as a first step towards building powerful third party tests of contamination, and we believe it is an exciting open problem to build tests that are capable of reliably detecting contamination at the single-duplication-count regime.

## 8 ACKNOWLEDGEMENTS

We gratefully acknowledge support from the CRFM Levanter team, especially David Hall, for both computational resources and infrastructure support, and to Google's TPU Research Cloud (TRC) for Cloud TPUs used in the pretraining experiments. Nicole Meister was supported by NSF GRFP DGE-2146755. Niladri Chatterji and Faisal Ladhak were supported by SAIL and Tatsunori Hashimoto was supported by a gift from IBM and a Hoffman-Yee grant. Finally, we would like to thank Nicholas Carlini and Percy Liang for insightful discussions on memorization and test set contamination.

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

APPENDICES

## A STRIDED LOG-LIKELIHOODS

To compute log-likelihoods for sequences exceeding the context length, we use a strided window approach, with a stride equal to half of the model's context length. We find that decreasing the stride beyond half the context length does not yield significant gains.

## B PRETRAINING DETAILS

We elaborate on the hyperparameters and training procedure of our 1.4B language model, trained from scratch on intentionally contaminated Wikitext.

We use a GPT-2 architecture with 1.4B parameters, with the architecture hyperparameters given by a hidden dimension of 1536, 24 heads, 48 layers, and a sequence length of 2048. The training batch size was 256. Based on the number of training tokens, sequence length, and training batch size, we trained this model for 46000 steps so as to consume the tokens in our mixture datasets exactly once. The model was optimized using AdamW with a learning rate of 1e-4 and weight decay of 0.1. We trained the model using Levanter on a v3-128 TPU instance on Google Cloud for 1.5 days (Hall et al. (2023)).

## C 10 CANARY DATASETS

In this section we provide additional details on the 10 canary datasets we injected into Wikitext to form our pretraining data. For BoolQ[3] (Clark et al., 2019), HellaSwag[4] (Zellers et al., 2019), MNLI[5] (Williams et al., 2018), Natural Questions[6] (Kwiatkowski et al., 2019a), TruthfulQA[7] (Lin et al., 2022), PIQA[8] (Bisk et al., 2019), we sample a random subset of 1000 examples. For Open-bookQA[9] (Mihaylov et al., 2018b), because of its smaller test set of size n=500, we used all 500 examples. Finally, for MMLU[10] (Hendrycks et al., 2021), we chose from subsets with no multi-line examples and having at least 500 examples, specifically Professional Psychology (n=611), MMLU Professional Law (n=1000), MMLU High School Psychology (n=544). Finally, we shuffle the examples in all datasets to make them exchangeable. In Table 3, we provide additional information about the injected datasets including number of examples, average words per example, and number of tokens per dataset. For each duplication rate, we included a short, medium and longer dataset, as measured by the total token count. The total token count of injected benchmarks is 19.235M tokens, meaning that the injected dataset is less than 0.1% of the entire pre-training dataset.

## D FULL MMLU RESULTS

We list in table 4 the full set of MMLU results used to generate the omnibus p-values for contamination by MMLU listed in table 2, before filtering out for suspected non-exchangeability.

## E COMPARISON TO MIN-K%-PROB BASELINE

We provide a comparison against Min-k% Prob (Shi et al. (2024)), a state of the art heuristic method for contamination detection proposed contemporaneous to our work. We use the same pretrained model and test sets from our experiments in Section 4.1.

Our implementation of Min-k%-Prob follows the methodology outlined in Shi et al. (2024); we run the method on one hundred 512-token spans sampled from each benchmark, and tune the decision

---

[3]`https://github.com/google-research-datasets/boolean-questions`
[4]`https://rowanzellers.com/hellaswag/`
[5]`https://cims.nyu.edu/~sbowman/multinli/`
[6]`https://github.com/google-research-datasets/natural-questions`
[7]`https://github.com/sylinrl/TruthfulQA/blob/main/data/finetune_truth.jsonl`
[8]`https://yonatanbisk.com/piqa/`
[9]`https://allenai.org/data/open-book-qa`
[10]`https://github.com/hendrycks/test`

Table 3: Injected canary datasets and duplication counts used in our pretraining experiments.

| Name | Examples | Avg Words/Ex | Tokens | Dup Rate (High) | Dup Rate (Low) |
|---|---|---|---|---|---|
| BoolQ | 1000 | 110 | 110k | 1 | 1 |
| HellaSwag | 1000 | 185 | 185k | 1 | 1 |
| OpenbookQA | 500 | 40 | 20k | 1 | 2 |
| Natural Questions | 1000 | 32 | 32k | 10 | 2 |
| MNLI | 1000 | 235 | 235k | 10 | 4 |
| TruthfulQA | 1000 | 25 | 25k | 10 | 4 |
| PIQA | 1000 | 50 | 50k | 50 | 7 |
| MMLU Pro. Law | 1000 | 2000 | 200k | 50 | 7 |
| MMLU Pro. Psych | 611 | 50 | 30k | 50 | – |
| MMLU H.S. Psych | 544 | 37 | 20k | 100 | – |

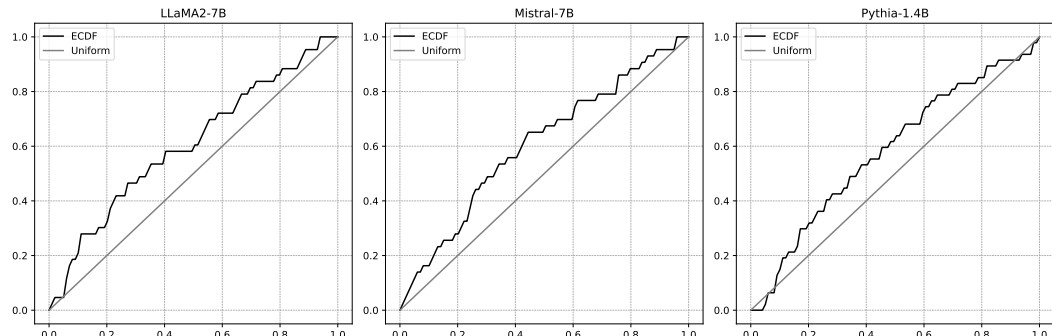

Figure 4: Empirical CDFs of MMLU p-values of LLaMA2, Mistral, and Pythia after exclusion of BioMedLM and GPT-2 significant test files, plotted against CDFs of a Uniform(0,1).

threshold on a validation set of five of our contaminated test sets, and five test sets not used in our data mixture. The threshold is tuned for a false positive rate of 5% to allow for a meaningful comparison against our test. A value of k=20 is used as is recommended in the paper.

Table 4: Full MMLU results for LLaMA2-7B, Mistral-7B, Pythia-1.4B, GPT-2XL and BioMedLM.

| Dataset | Size | LLaMA2-7B | Mistral-7B | Pythia-1.4B | GPT-2 XL | BioMedLM |
|---------|------|-----------|-----------|-------------|----------|----------|
| Abstract-Algebra | 100 | 0.103 | 0.246 | 0.645 | 0.894 | 0.861 |
| Anatomy | 135 | 0.586 | 0.605 | 0.605 | 0.439 | 0.649 |
| Astronomy | 152 | 0.550 | 0.657 | 0.050 | 0.003 | 0.050 |
| Business-Ethics | 100 | 0.936 | 0.440 | 0.107 | 0.808 | 0.499 |
| Clinical-Knowledge | 265 | 0.199 | 0.108 | 0.081 | 0.004 | 0.268 |
| College-Biology | 144 | 0.108 | 0.051 | 0.152 | 0.260 | 0.779 |
| College-Chemistry | 100 | 0.526 | 0.614 | 0.595 | 0.355 | 0.265 |
| College-Computer-Science | 100 | 0.060 | 0.498 | 0.383 | 0.532 | 0.267 |
| College-Mathematics | 100 | 0.059 | 0.151 | 0.162 | 0.321 | 0.727 |
| College-Medicine | 173 | 0.397 | 0.106 | 0.340 | 0.440 | 0.067 |
| College-Physics | 102 | 0.694 | 0.757 | 0.972 | 0.719 | 0.262 |
| Computer-Security | 100 | 0.214 | 0.007 | 0.314 | 0.180 | 0.928 |
| Conceptual-Physics | 235 | 0.554 | 0.333 | 0.811 | 0.710 | 0.924 |
| Econometrics | 114 | 0.616 | 0.761 | 0.540 | 0.508 | 0.035 |
| Electrical-Engineering | 145 | 0.266 | 0.364 | 0.595 | 0.490 | 0.277 |
| Elementary-Mathematics | 378 | 0.059 | 0.416 | 0.260 | 0.355 | 0.528 |
| Formal-Logic | 126 | 0.666 | 0.750 | 0.990 | 0.930 | 0.398 |
| Global-Facts | 100 | 0.779 | 0.957 | 0.448 | 0.339 | 0.667 |
| High-School-Biology | 310 | 0.645 | 0.279 | 0.476 | 0.499 | 0.416 |
| High-School-Chemistry | 203 | 0.229 | 0.426 | 0.813 | 0.539 | 0.055 |
| High-School-Computer-Science | 100 | 0.023 | 0.085 | 0.106 | 0.045 | 0.484 |
| High-School-European-History | 165 | 0.009 | 1e-38 | 1e-38 | 1e-38 | 1e-38 |
| High-School-Geography | 198 | 0.194 | 0.339 | 0.341 | 0.721 | 0.277 |
| High-School-Government-And-Politics | 193 | 0.294 | 0.066 | 0.025 | 0.372 | 0.003 |
| High-School-Macroeconomics | 390 | 0.543 | 0.222 | 0.276 | 0.236 | 0.511 |
| High-School-Mathematics | 270 | 0.473 | 0.182 | 0.001 | 0.272 | 0.032 |
| High-School-Microeconomics | 238 | 0.862 | 0.797 | 0.712 | 0.122 | 0.181 |
| High-School-Physics | 151 | 0.339 | 0.757 | 0.171 | 0.114 | 0.354 |
| High-School-Psychology | 545 | 0.033 | 0.037 | 0.004 | 0.039 | 0.007 |
| High-School-Statistics | 216 | 0.077 | 0.119 | 0.228 | 0.233 | 0.211 |
| High-School-Us-History | 204 | 1e-38 | 1e-38 | 1e-38 | 1e-38 | 0.001 |
| High-School-World-History | 237 | 1e-38 | 1e-38 | 1e-38 | 1e-38 | 1e-38 |
| Human-Aging | 223 | 0.882 | 0.830 | 0.617 | 0.677 | 0.803 |
| Human-Sexuality | 131 | 0.807 | 0.959 | 0.502 | 0.789 | 0.471 |
| International-Law | 121 | 0.061 | 0.545 | 0.532 | 0.582 | 0.817 |
| Jurisprudence | 108 | 0.009 | 0.075 | 0.852 | 0.879 | 0.616 |
| Logical-Fallacies | 163 | 0.103 | 0.188 | 0.082 | 0.397 | 0.193 |
| Machine-Learning | 112 | 0.266 | 0.297 | 0.373 | 0.061 | 0.868 |
| Management | 103 | 0.516 | 0.209 | 0.454 | 0.907 | 0.281 |
| Marketing | 234 | 0.874 | 0.684 | 0.977 | 0.848 | 0.451 |
| Medical-Genetics | 100 | 0.501 | 0.037 | 0.523 | 0.425 | 0.729 |
| Miscellaneous | 783 | 0.099 | 0.122 | 0.086 | 0.266 | 0.081 |
| Moral-Disputes | 346 | 0.017 | 0.011 | 0.097 | 0.190 | 0.157 |
| Moral-Scenarios | 895 | 0.652 | 0.022 | 0.121 | 0.487 | 0.413 |
| Nutrition | 306 | 0.163 | 0.601 | 0.413 | 0.865 | 0.609 |
| Philosophy | 311 | 0.011 | 0.094 | 0.003 | 0.049 | 0.044 |
| Prehistory | 324 | 0.203 | 0.412 | 0.055 | 0.268 | 0.262 |
| Professional-Accounting | 282 | 0.708 | 0.242 | 0.197 | 0.910 | 0.798 |
| Professional-Medicine | 272 | 0.001 | 0.017 | 0.001 | 0.002 | 1e-38 |
| Professional-Psychology | 612 | 0.001 | 0.001 | 0.058 | 0.009 | 0.201 |
| Public-Relations | 110 | 0.402 | 0.237 | 0.162 | 0.512 | 0.622 |
| Security-Studies | 245 | 0.070 | 0.043 | 0.687 | 0.061 | 0.073 |
| Sociology | 201 | 0.350 | 0.258 | 0.260 | 0.660 | 0.496 |
| Us-Foreign-Policy | 100 | 0.934 | 0.252 | 0.778 | 0.646 | 0.173 |
| Virology | 166 | 0.203 | 0.854 | 0.219 | 0.796 | 0.212 |
| World-Religions | 171 | 0.311 | 0.882 | 0.933 | 0.222 | 0.851 |

Table 5: Canary results for Min-k%-Prob.

| Name | Size | Dup Count | Min-k% Prob |
|---|---|---|---|
| BoolQ | 1000 | 1 | 13% |
| HellaSwag | 1000 | 1 | 2% |
| MNLI | 1000 | 10 | 100% |
| MMLU Pro. Law | 1533 | 50 | 100% |
| MMLU H.S. Psychology | 544 | 100 | 100% |

