# OpenReview forum: "Proving Test Set Contamination in Black-Box Language Models"
_ICLR.cc/2024/Conference — ICLR 2024 oral_

### Official Review · Reviewer_FS83 · 2023-10-30

**Soundness:** 2 fair
**Presentation:** 3 good
**Contribution:** 2 fair
**Rating:** 6
**Confidence:** 3

**Summary:**

This paper targets the problem of detecting test set contamination of black-box language models. The proposed method is based on two hypotheses: (1) the exchangeability of many datasets (distribution won't be affected after shuffling); and (2) if a language model is contaminated, it is more likely to find certain orderings of data samples than other orderings. Then a statistical test is proposed to compare the log probability of the dataset under the original ordering to the log probability under random permutations on sharded datasets. Experiments are conducted with one 1.4B-gpt2 model trained from scratch on 10 test sets, and the results prove the effectiveness of the proposed framework.

**Strengths:**

- This paper targets an interesting and exciting problem in the community, test set contamination.
- Based on the hypothesis, this paper proposed a contamination detection method, which is intuitive and easy to deploy in other settings.
- The method is verified with a 1.4B language model trained from scratch, and the existing Llama2 model, both showing promising results even when the test set only appears a few times in the pre-training corpus.

**Weaknesses:**

- I'm most concerned about the definition of contamination used in this paper. Currently, the most popular definition of contamination follows the n-gram analysis. In real-world scenarios when training large language models, it's hardly seen to directly feed original data samples in their original ordering as shown in Figure 1. The application of this work could be greatly limited.
- From Figure 3, it seems that the parameters for shards and permutations are sensitive and have to be carefully selected when being applied to other test sets.
- The paper only targets direct sentence appearance in the pre-training stage. What about instruction-tuning data in the SFT stage?

**Questions:**

- Could you further explain "high false positives" in existing n-gram-based analyses?
- How did you deal with the labels for the data samples in test sets?

---

> ### Author Response · Authors · 2023-11-17
>
> Thank you for your thorough analysis and constructive feedback on our paper. We appreciate the opportunity to clarify the points raised and to provide additional insights into our research.
>
> N-gram overlap is a commonly utilized measure of contamination in the literature; however, it should be noted that it acts more as a measurement tool rather than as a definition of contamination. N-gram overlap may fail to distinguish between coincidental overlap and genuine contamination in some situations, and has been observed to potentially lead to false positives under certain conditions. For example, the Stanford Question-Answering Dataset (SQuAD v2) uses background information derived from Wikipedia, and in the GPT-3 paper, Brown et. al. (2020) find high N-gram overlap for this reason, even if labels are not present in the data and no true contamination exists. Please see pg. 43 of that paper for examples of false positives in their contamination study.
>
> We believe that the definition introduced in our paper, of contamination as a statistical dependence between the model and the dataset, is precise and formal, and better captures the notion of contamination as a transfer of information between the test set and the model—not simply a correlation that appears because both the test set and pre-training data share information.
>
> While verbatim contamination of ordered data does not encompass all forms of contamination, we found that the presence of ordered test sets in pre-training data is surprisingly common. A search of The Pile, a large open-source language modeling dataset, yielded numerous instances of real-world datasets embedded with examples appearing in-order; see our top-level comment for an example.
>
> In pre-training, shuffling of the data occurs at the document level, and is not typically applied to the data within a document. Files collected from the internet would typically be treated as singular documents in dataset construction pipelines.
>
> More importantly, the use of ordering allows us to give provable guarantees of contamination, which is more difficult to achieve for other, less direct forms of contamination. Our work is the only existing contamination detection method for language models to give guarantees of this kind.
>
> Regarding concerns about sensitivity of the test to the shard count, figure 3 (left) shows that a wide range of shard counts (between 10 and 150) attain p-values below 1e-4. Once the p-value is low enough that statistical significance is attained, there is no added benefit to lowering the p-value. Therefore, the plot suggests that the test is robust to shard count, so long as the shard count is not too low (so that the t-test can still be used reasonably), and not too high (so that there are enough examples per shard to get sufficient signal from log-prob differences.)
>
> Similarly, figure 3 (right) shows that increasing the permutation count monotonically decreases the p-value, and that the p-value stabilizes beyond about 25 permutations per shard. This suggests that the test is not sensitive to the permutation count, provided that the permutation count is not too low. Our empirical results use a permutation count of at least 50. We welcome further discussion on this point to ensure we fully understand and address your concerns.
>
> Detecting contamination at the instruction fine-tuning stage would be interesting follow-up work, but is complicated by the fact that examples are commonly shuffled in this setting, and so we cannot test against a known example order. In this setting, heuristic methods may prove to be more effective.
>
> We hope this response has addressed your concerns effectively. We are grateful for the chance to discuss our work's potential, and wish to thank you again for your valuable input.

---

### Official Review · Reviewer_gkwi · 2023-11-01

**Soundness:** 3 good
**Presentation:** 3 good
**Contribution:** 3 good
**Rating:** 8
**Confidence:** 3

**Summary:**

This paper examines the issue of test set contamination in large language models (LLMs), referring to the phenomenon where LLMs memorize public benchmarks during their pretraining phase. Since the pretraining datasets are rarely available, this paper proposes a statistical test to identify the presence of a benchmark in the pre-training dataset of a language model without accessing the model’s training data or weights. The intuition is the exchangeability of datasets— the order of examples in the dataset can be shuffled without affecting its joint distribution. If a language model shows a preference for any ordering of the dataset, it might have seen the data during pretraining. The test on the LLaMA-2 model identifies potential contamination in the MMLU benchmark, which is consistent with the results in the original LLaMA-2 report.

**Strengths:**

-	The idea of utilizing dataset exchangeability to identify test set contamination is novel and interesting.
-	The proposed sharded likelihood comparison test addresses the tradeoff between statistical power and computational requirements of the permutation test, which is promising. The sharded rank comparison test also provides (asymptotic) guarantees on false positive rates.
-	Experimental results are promising. A GPT-2 model is trained from scratch on standard pretraining data and known test sets to verify the efficiency of the proposed method in identifying test set contamination. The method is also tested with an existing model, LLaMA2, on the MMLU dataset, showing general agreement with the contamination study results.

**Weaknesses:**

-	Although a more efficient sharded rank comparison test is proposed, the computational complexity is still considerable. For example, testing 49 files using 1000 permutations per shard can take 12 hours for LLaMA2.
-	There is no comparison with other baseline methods.
-	The method relies on a strong assumption of data exchangeability, which may not hold in real-world datasets.

**Questions:**

If a dataset is not exchangeable, how effective is the method?

---

> ### Author Response · Authors · 2023-11-17
>
> Thank you for your thorough review and valuable feedback on our work.
>
> We'd like to address the concern regarding the computational complexity of our test. It's important to note that the test is a one-time process for any given model and dataset; once the p-values are computed, there is no need for recalculation. Our findings indicate that a number of permutations beyond 30-50 per shard offers diminishing returns, as shown in Figure 3 (right).
>
> Furthermore, the test's design allows for easy parallelization. Each shard permutation can be evaluated independently, enabling the use of inexpensive commodity hardware to run the test significantly faster.
>
> Regarding the assumption of data exchangeability, this is a strictly weaker condition than the commonly held assumption of independent and identically distributed (I.I.D.) data in machine learning. Most datasets satisfy this assumption to some extent.
>
> We acknowledge the validity of our test hinges on data exchangeability. However, depending on the source of non-exchangeability, it is often the case that a dataset can be altered slightly so that our test is still valid. For example, a common source of non-exchangeability is the presence of ascending IDs (e.g. as in SQuAD and HumanEval). We can adjust the data—by either removing these IDs or permuting the examples while keeping IDs constant—to retain the test's applicability. This is discussed in more detail in the revised paper.
>
> Finally, we appreciate your suggestion to include baseline comparisons. We provide a comparison against a contamination detection method called Min-K% Prob, a state of the art heuristic method for contamination detection in language models proposed contemporaneous to our work by Shi et. al. (2023).
>
> We find that our method matches or exceeds the performance of this state of the art heuristic method. Please see the table in the top-level comment for numbers.

---

> > ### Comment · Reviewer_gkwi · 2023-11-22
> >
> > I appreciate the authors' response, which addresses most of my concerns. I am happy to raise my rating.

---

### Official Review · Reviewer_Rgfh · 2023-11-01

**Soundness:** 3 good
**Presentation:** 4 excellent
**Contribution:** 4 excellent
**Rating:** 8
**Confidence:** 3

**Summary:**

This paper studies the problem of identifying test set contamination in large language models, i.e., detecting that a test set is present in the pretraining data of a language model. The main idea behind the approach is that for test sets that have some canonical order of individual instances (e.g.: the order in which the dataset creators release the dataset), the likelihood of the test set in that order would be significantly higher than any random permutation of the dataset. Based on this idea, the paper proposes two versions of the test, one of which shards the test set and aggregates statistics over the shards to make the estimate more robust to potential biases in the model.

The tests are evaluated first by measuring their sensitivity when pretraining datasets are intentionally contaminated. It is shown that they are highly sensitive when the tests sets are large or have been duplicated enough in the pretraining data. The test is then used to measure contamination of the pretraining data used to train the Llama models and it is shown that the findings agree with prior reports.

**Strengths:**

This is clearly written paper and makes a strong contribution. The tests do not require access to model weights or pretraining data, making them practically useful.

**Weaknesses:**

The experiments do not compare the performance of the proposed tests to prior work. I understand that this work differs from say, the work from Carlini et al. in that this work focuses on set-level contamination, but how does aggregating instance-level statistics over a set compare?

**Questions:**

- How does the performance of this method compare to that of prior work (see Weakness)
- How sensitive is the proposed test to the model size?

---

> ### Author Response · Authors · 2023-11-17
>
> Thank you for your recognition of our work and for your insightful comments.
>
> This work differs from prior work in two key ways: in the definition and setting considered and in the resultant provable guarantees. We consider contamination detection as the problem of detecting a statistical dependence between the test data and model parameters, and show that we can provide provable guarantees in the case of verbatim contamination, where the full test set (with examples and labels) is embedded in the pretraining data. Prior work is primarily heuristic in nature; To our knowledge, our work is the first of its kind to provide provable guarantees of contamination for language models.
>
> For comparison against a baseline, we provide a comparison against a contamination detection method called Min-K% Prob, a state of the art heuristic method for contamination detection in language models proposed contemporaneous to our work by Shi et. al. (2023).
>
> We find that our method matches or exceeds the performance of this state of the art heuristic method. Please see the table in the top-level comment for numbers.
>
> Thank you for your question regarding the impact of model size on the performance of the test. Existing work on memorization in language models suggests that larger models memorize their training data more strongly. For example, Carlini et. al. (2023) show that both model size and repetitions in the training data increase the extractability of training data sequences. Our empirical results show that the power of our test increases dramatically in the number of repetitions (duplication rate), and we posit that our test enjoys a similar increase in power for larger model sizes, since larger models likely memorize example order more strongly.
>
> We present preliminary results on the impact of model size on the power of our test. We evaluated three models of increasing parameter counts trained on the same data mixtures as in section 4.1, on the test sets present in the pretraining data at a duplication rate of 1. These results suggest that the test performs better with larger models.
>
> | Parameter Count | Average Logarithm of P-Value |
> |-----------------|---------|
> | 355M          | -1.427  |
> | 774M          | -1.825 |
> | 1500M          | -12.783  |

---

### Official Review · Reviewer_jRLi · 2023-11-04

**Soundness:** 3 good
**Presentation:** 3 good
**Contribution:** 3 good
**Rating:** 8
**Confidence:** 3

**Summary:**

The paper proposes a statistical test that given certain assumptions can indicate whether a black-box language model has been trained on certain datasets. This is a topic of increasing interest and importance given the prevalence of pretrained models that are trained on very large amounts of data.  The authors first propose a simple permutation test and identify some weaknesses with it. They then propose a more sophisticated sharded test. The authors show 2 kinds of experiments:

(1) They test on a dataset where they have injected a small amount of certain test sets to see if their approach can detect them.

(2) They apply their test to existing models such as Lllama-2 showing their approach can scale.

**Strengths:**

-Topic of large importance in the community given the direction of the field.

-Novel approach with thorough empirical results. I have some questions about the definition of test set contamination below.

-Well written and interesting.

**Weaknesses:**

I have some questions about the definition of test set contamination below.

**Questions:**

In Figure 1 the authors show test set contamination for BoolQ. But the examples there are unlabeled. Are the authors targeting unlabeled test set contamination i.e. the input is present in the pretraining data but not the label?

Would be great to have some justification and explanation of this setting.

---

> ### Author Response · Authors · 2023-11-17
>
> Thank you for your thoughtful review and for recognizing the importance of our work.
>
> Regarding the definition of test set contamination, we specifically address verbatim contamination, where both inputs and labels from the test set appear in the training data in order. While other forms of contamination can be studied, such as indirect contamination, we study verbatim contamination as this form of contamination is more amenable to provable detection with statistical guarantees.
>
> Your second question is regarding the omission of labels In Figure 1. We omitted labels in the figure for readability, and the actual test does incorporate the labels.

---

### Author Response · Authors · 2023-11-17

We are sincerely grateful to the reviewers for dedicating their time and effort to review our work, and we appreciate the recognition of the novelty of using exchangeability for contamination detection and the significance of our contribution given the discourse surrounding contamination in the field. We address each reviewer's comments in detail below. We have made numerous updates to the submission, most notably with the results of our test on four popular open models and eight commonly used benchmarks.

One question shared by multiple reviewers is regarding the exact notion of contamination we consider in this work. Rather than consider a definition based on heuristics like n-gram overlap, we consider contamination detection as the problem of detecting statistical dependence between the test data and model parameters. Within this setting, our work shows that it is possible to provide provable guarantees of contamination in the case of verbatim contamination, where the full test set (with examples and labels) is embedded in the pretraining data.

To illustrate the relevance of this setting, we note that a search of The Pile, a large open-source language modeling dataset, yielded numerous instances of small real-world datasets embedded with examples appearing in-order. As one example, the following is an excerpt from a dataset for an annotation tool made by Explosion, the creators of spaCy, a popular natural language processing framework, found in The Pile:

```
{"text":"Uber\u2019s Lesson: Silicon Valley\u2019s Start-Up Machine Needs Fixing","meta":{"source":"The New York Times"}}
{"text":"Pearl Automation, Founded by Apple Veterans, Shuts Down","meta":{"source":"The New York Times"}}
{"text":"How Silicon Valley Pushed Coding Into American Classrooms","meta":{"source":"The New York Times"}}

Source: https://github.com/explosion/prodigy-recipes/tree/fc06f6a6d93bc477e98cf0d8357c39322e4f5a6a
```

What our work shows is that by exploiting exchangeability in this setting, we are able to provide guarantees on the false positive rate of our test.

Multiple reviewers indicated the desire for a comparison against a baseline method. While no other existing work is comparable in the sense that it provides a statistical proof of contamination like ours, we provide a comparison against a state of the art heuristic method for contamination detection called Min-K% Prob, proposed by Shi et. al. (2023) contemporaneous to our work. We use the same pretrained model and test sets from our experiments in Section 4.1.

| Dataset    | Duplication Count | Sharded p (ours) | Percent Contaminated (Min-K%-Prob) |
|------------|-------------------|------------------|------------------------------------|
| BoolQ     | 1            | 0.156          | 3%                             |
| HellaSwag | 1            | 0.478          | 2%                            |
| MNLI     | 10            | 1.96e-11           | 100%                           |
| MMLU-Pro-Law | 50        |  1e-38           | 90%                         |
| MMLU-HS-Psych | 100     |  1e-38           | 74% |

Our run of Min-k%-Prob follows the methodology outlined in the paper; we run the method on one hundred 512-token spans sampled from each benchmark, and tune the decision threshold on a validation set of five of our contaminated test sets, and five test sets not used in our data mixture (uncontaminated). The threshold is tuned for a false positive rate of 5% to allow for a meaningful comparison against our test. A value of k=20 is used as is recommended in the paper.

We find that our method matches or exceeds the performance of this state of the art heuristic method, while also providing statistical proof of contamination.

---

### Meta-Review · Area_Chair_yTmG · 2023-12-13

**Metareview:**

This paper studies the problem of identifying test set contamination in large language models, i.e., detecting that a test set is present in the pretraining data of a language model. The main idea behind the approach is that for test sets that have some canonical order of individual instances (e.g.: the order in which the dataset creators release the dataset), the likelihood of the test set in that order would be significantly higher than any random permutation of the dataset.
The paper is well written, relevant and interesting. The approach is novel and well defined and does work for blackbox LLMs. The missing comparison to other work has been added in the author response and should address all concerns of all reviewers.

**Justification For Why Not Higher Score:**

N/A

**Justification For Why Not Lower Score:**

This is a super relevant paper and the methods are novel and well defined. This is a great paper.

---

### Decision · Program_Chairs · 2024-01-16

Accept (oral)